# Sparse VideoGen: Accelerating Video Diffusion Transformers with Spatial-Temporal Sparsity

**Haocheng Xi** [*] **Shuo Yang** [*] **Yilong Zhao Chenfeng Xu Muyang Li Xiuyu Li Yujun Lin Han Cai**
**Jintao Zhang Dacheng Li Jianfei Chen Ion Stoica Kurt Keutzer Song Han**

https://github.com/svg-project/Sparse-VideoGen

## Abstract

Diffusion Transformers (DiTs) dominate video generation but their high computational cost severely limits real-world applicability, usually requiring tens of minutes to generate a few seconds of video even on high-performance GPUs. This inefficiency primarily arises from the quadratic computational complexity of 3D full attention with respect to the context length. In this paper, we propose a training-free framework termed Sparse VideoGen (**SVG**) that leverages the inherent sparsity in 3D full attention to boost inference efficiency. We reveal that the attention heads can be dynamically classified into two groups depending on distinct sparse patterns: (1) *Spatial Head*, where only spatially-related tokens within each frame dominate the attention output, and (2) *Temporal Head*, where only temporally-related tokens across different frames dominate. Based on this insight, SVG proposes an online profiling strategy to capture the dynamic sparse patterns and predicts the type of attention head. Combined with a novel hardware-efficient tensor layout transformation and customized kernel implementations, SVG achieves up to $2.28\times$ end-to-end speedup on CogVideoX-v1.5, $2.33\times$ on Hunyuan-Video, while preserving generation quality. Our code is open-sourced at https://github.com/svg-project/Sparse-VideoGen.

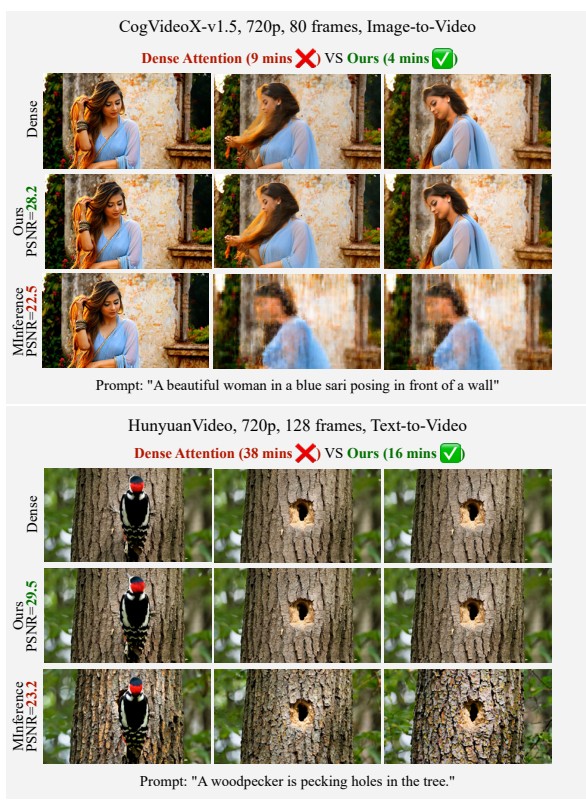

*Figure 1.* **SVG accelerates video generation while maintaining high quality**. On CogVideoX-v1.5-I2V and Hunyuan-T2V, our method achieves a $2.28\times$ and $2.33\times$ speedup with high PSNR. In contrast, MInference (Jiang et al., 2024) fails to maintain pixel fidelity (significant blurring in the first example) and temporal coherence (inconsistencies in the tree trunk in the second example).

## 1. Introduction

Diffusion Transformers (DiTs) (Peebles & Xie, 2023) have recently emerged as a transformative paradigm for generative tasks, achieving state-of-the-art results in image generation. This success has been naturally carried over to video generation, with models adapting from a spatial 2D attention to a spatiotemporal 3D full attention (Arnab et al., 2021;

[*]Equal contribution. University of California, Berkeley. NVIDIA. MIT. Correspondence to: Chenfeng Xu <xuchenfeng@berkeley.edu>.

*Proceedings of the 42nd International Conference on Machine Learning*, Vancouver, Canada. PMLR 267, 2025. Copyright 2025 by the author(s).

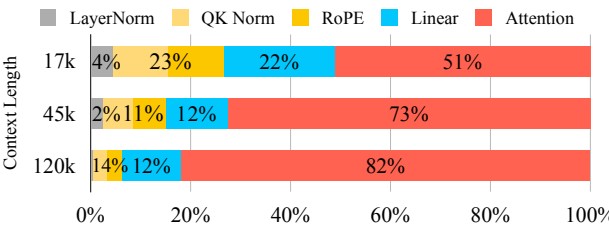

*Figure 2.* **Attention dominates the computation in video diffusion models.** For CogVideoX-v1 and -v1.5 with 17k and 45k context length, attention takes 51% and 73% of the latency, respectively. For HunyuanVideo with 120k context length, attention can take over 80% amount of the runtime latency.

Yang et al., 2024c; Kong et al., 2024), resulting in high-fidelity and temporally consistent outputs. Close-sourced models such as Sora and Kling, and open-sourced models including Wan 2.1 (Wang et al., 2025), CogVideo (Hong et al., 2023), and HunyuanVideo (Kong et al., 2024), have showcased impressive capabilities in applications ranging from animation (Guo et al., 2024; Feng et al., 2024) to physical world simulation (Liu et al., 2024b).

Despite significant advances in generating high-quality videos, the deployment of video generation models remains challenging due to their substantial computation usage. For instance, HunyuanVideo requires almost an hour on an NVIDIA A100 GPU to generate only a 5-second video, where the 3D full attention accounts for more than 80% of end-to-end runtime (Figure 2). Moreover, due to the *quadratic computational complexity* with respect to the context length (Dao et al., 2022), the attention can be much more dominant as the resolution and number of frames increase, as shown in Figure 2.

Fortunately, attention in transformers is well-known for its sparsity, offering a great opportunity to reduce redundant computation. For example, in large language models (LLMs), a small portion of the tokens can dominate the attention output (Zhang et al., 2023c; Xiao et al., 2024b; Tang et al., 2024). Therefore, the computation can be dramatically reduced by only computing the attention among such important tokens, while still maintaining generation accuracy. However, existing methods cannot be directly applied to DiTs (as shown in Table 1), as video data has fundamentally different sparsity patterns from text data.

Our key observation is that attention heads in DiTs exhibit inherent sparsity in two categories: *Spatial Head* and *Temporal Head*, based on their distinct sparsity patterns. As shown in Figure 3, spatial head mainly focuses on tokens that reside within the same frame, which determines the spatial structures of generated videos. In contrast, temporal head attends to tokens at the same spatial location across all frames, contributing to the temporal consistency. There-

fore, the computation for both types of heads can be greatly reduced by only calculating the attended tokens.

Despite the theoretical speedup, leveraging sparsity for end-to-end acceleration is still challenging. Firstly, sparsity patterns are highly dynamic across different denoising steps and input prompts. It necessitates an online method to identify sparsity patterns without incurring overhead. Secondly, some sparsity patterns are unfriendly to hardware accelerators. For example, temporal head computes over noncontiguous data that cannot be fed to GPU's tensor cores, resulting in significant efficiency degradations (Ye et al., 2023).

To tackle these challenges, we propose Sparse VideoGen (SVG), a training-free framework that accelerates video DiTs with the following novel designs: (1) To efficiently identify the best sparsity pattern for each attention head, SVG introduces an online profiling strategy with minimal overhead (∼3%). It randomly samples 1% tokens from each attention head and processes sampled tokens with full attention computation and two distinct sparse attentions (spatial head and temporal head). Finally, the sparse pattern with a lower error compared to the full attention is selected for each head. (2) To improve hardware efficiency, SVG proposes a novel layout transformation, which reorders the noncontiguous sparsity pattern of temporal head into a compact and hardware-friendly sparsity pattern.

We prototype SVG with customized kernel implementation by Triton (Tillet et al., 2019) and FlashInfer (Ye et al., 2025) and evaluate SVG's accuracy and efficiency on representative open video generative models including CogVideoX-v1.5-I2V, CogVideoX-v1.5-T2V, and HunyuanVideo-T2V. SVG delivers significant efficiency improvements, achieving an end-to-end speedup of up to 2.33× while maintaining high visual quality with a PSNR of up to 29, outperforming all prior methods. Additionally, we show that SVG is compatible with FP8 quantization, enabling additional efficiency gains without compromising quality. We summarize our key contributions as follows:

- We conduct in-depth analysis of video DiTs' sparse patterns, revealing two inherent sparse attention patterns (temporal head and spatial head) for efficient video generation.

- We propose SVG, a training-free sparse attention framework comprising an efficient online profiling strategy and an efficient inference system for accurate and efficient video generation.

- SVG delivers significant speedup while maintaining good video generation quality, paving the way for practical applications of video generative models.

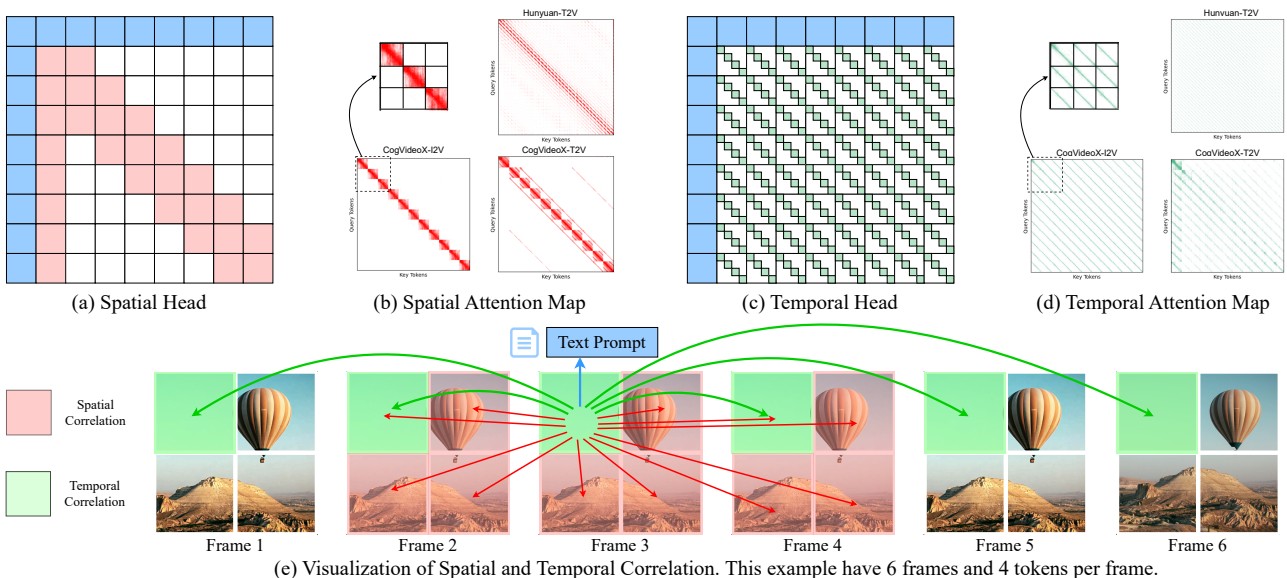

(a) Spatial Head     (b) Spatial Attention Map     (c) Temporal Head     (d) Temporal Attention Map

(e) Visualization of Spatial and Temporal Correlation. This example have 6 frames and 4 tokens per frame.

*Figure 3.* We observe two types of attention maps with distinct sparse patterns: *spatial map (b)* and *temporal map (d)*. Based on the attention map, we classify all attention heads into *Spatial Head (a)* and *Temporal Head (c)*, which contribute to the spatial and temporal consistency of generated videos respectively. As visualized in (e), spatial head primarily focuses on all tokens within the same frame (painted as red). In contrast, temporal head attends to tokens at the same position across all frames (painted as green).

## 2. Related Work

### 2.1. Efficient diffusion models

**Decreasing the denoising steps.** Most diffusion models employ SDEs that require many sampling steps (Song & Ermon, 2019; Ho et al., 2020; Meng et al., 2022). To address this, DDIM (Song et al., 2020) approximates them with an ODE; subsequent techniques refine ODE paths and solvers (Lu et al., 2022a;b; Liu et al., 2022; 2024c) or incorporate consistency losses (Song et al., 2023; Luo et al., 2023). Distillation-based methods (Yin et al., 2024a;b) train simpler, few-step models. However, these require expensive re-training or fine-tuning—impractical for most video use cases. In contrast, our approach directly uses off-the-shelf pre-trained models without any additional training.

**Diffusion model compression.** Weight compression through quantization is a common tactic (Li et al., 2023; Zhao et al., 2024a; Li* et al., 2025), pushing attention modules to INT8 (Zhang et al., 2025b) or even INT4/FP8 (Zhang et al., 2024). Other work proposes efficient architectures (Xie et al., 2024; Cai et al., 2024; Chen et al., 2025) or high-compression autoencoders (Chen et al., 2024a) to improve performance. Our Sparse VideoGen is orthogonal to these techniques and can incorporate them for additional gains.

**Efficient system implementation.** System-level optimizations focus on dynamic batching (Kodaira et al., 2023; Liang et al., 2024), caching (Chen et al., 2024b; Zhao et al., 2024b), or hybrid strategies (Lv et al., 2024; Liu et al., 2024a). While these methods can improve throughput, their output quality often drops below a PSNR of 22. By contrast, our method preserves a PSNR above 30, thus substantially outperforming previous approaches in maintaining output fidelity.

### 2.2. Efficient attention methods

**Sparse attention in LLMs.** Recent research on sparse attention in language models reveals diverse patterns to reduce computational overhead. StreamingLLM (Xiao et al., 2023) and LM-Infinite (Han et al., 2023) observe that attention scores often concentrate on the first few or local tokens, highlighting temporal locality. H2O (Zhang et al., 2023b), Scissorhands (Liu et al., 2024d) and DoubleSparsity (Yang et al., 2024b) identify a small set of "heavy hitter" tokens dominating overall attention scores. TidalDecode (Yang et al., 2024a) shows that attention patterns across layers are highly correlated, while DuoAttention (Xiao et al., 2024a) and MInference (Jiang et al., 2024) demonstrate distinct sparse patterns across different attention heads. However, these methods focus on token-level sparsity and do not leverage the inherent redundancy of video data.

**Linear and low-bit attention.** Another direction involves linear attention (Cai et al., 2023; Xie et al., 2024; Wang et al., 2020; Choromanski et al., 2020; Yu et al., 2022; Katharopoulos et al., 2020), which lowers complexity from quadratic to linear, and low-bit attention (Zhang et al., 2025b; 2024), which operates in reduced precision to accelerate attention module. Sparse VideoGen is orthogonal to both approaches:

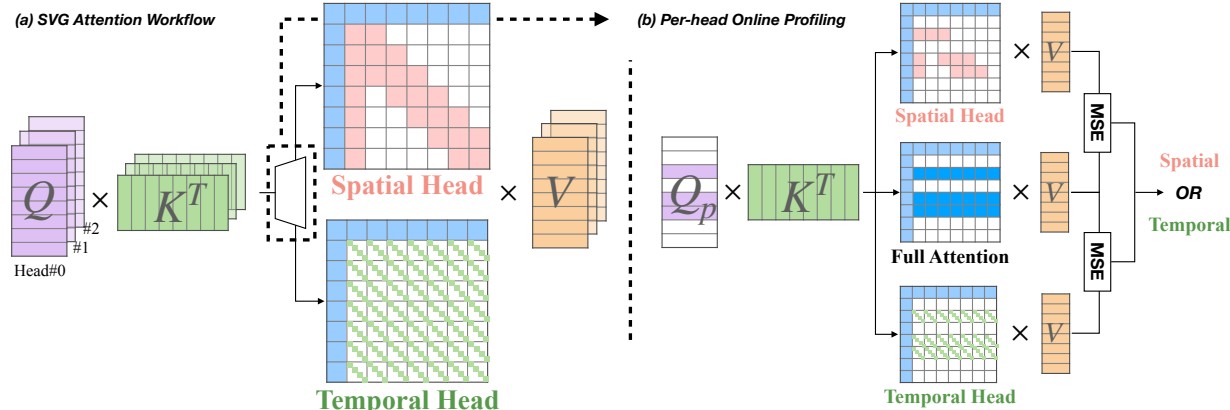

*Figure 4.* Overview of SVG framework. (a) During generation, SVG adaptively classifies each attention head as either a *spatial head* or a *temporal head* and applies a dedicated sparse attention computation accordingly. (b) This adaptive classification is driven by online profiling strategy, which extracts a small portion of $Q$, denoted as $Q_p$, to perform both spatial and temporal attention computations. SVG then selects the attention patter that yields the minimal MSE compared to full attention, ensuring accurate classification.

it can be combined with techniques like FP8 attention while still benefiting from the video-specific spatial and temporal sparsity in video diffusion models.

## 3. Motivation and Analysis

### 3.1. 3D Full Attention shows instinct sparsity

We identify that 3D full attention possess inherent sparsity, characterized by different distinct sparse patterns tailored for different functions (Xiao et al., 2024a). We deeply investigate the sparsity nature across various text-to-video and image-to-video models and identify two types of attention heads based on sparse patterns: *Spatial Head* and *Temporal Head*, as shown in Figure. 3.

**Spatial Head.** As illustrated in Figure 3(a-b), spatial head primarily focuses its attention scores on spatially-local tokens. This leads to the attention map exhibiting a *block-wise layout*. Since pixels within the same frame are tokenized into contiguous tokens, spatial head attends exclusively to pixels within the same frame and its adjacent frames. This property is essential for maintaining the spatial consistency of generated videos. In spatial head, the block size relates to the *number of tokens per frame*.

**Temporal Head.** In contrast, temporal head exhibits a *slash-wise layout* with a constant interval (Figure 3(c-d)). Since each frame is tokenized into a fixed number of tokens $L$, pixels occupying the same spatial position across different frames are arranged at a stride of $L$. Consequently, temporal head captures information from the token with the same spatial position across multiple frames. This pattern is important for ensuring temporal consistency in video generation[1].

---

[1]We hypothesize that this occurs because the majority of the

In addition to the spatial and temporal patterns, we observe that the text prompts and the first frame hold significant attention scores for both spatial and temporal head, which aligns with previous investigations (Xiao et al., 2024b; Shen et al., 2024; Su et al., 2025). Therefore, we include these tokens in both the spatial and temporal head.

### 3.2. Sparse attention achieves lossless accuracy

Furthermore, we find that directly applying sparse patterns to corresponding heads does not hurt the quality of generated videos. We demonstrate this by evaluating CogVideoX-v1.5 and HunyuanVideo on VBench (Huang et al., 2023) with sparse attention. We determine the sparse pattern by computing full attention along with two different sparse mechanisms (spatial head and temporal head) for each attention head and denoising step. The sparse pattern with the lowest mean squared error (MSE) relative to full attention is chosen for further inference. This approach achieves a PSNR over 29, showing that the right sparse pattern maintains the high quality of generated videos.

However, despite its accuracy, this strategy does not provide practical efficiency benefits, as full attention computation is still required to determine the optimal sparse pattern. We will address this issue in Sec 4.1.

### 3.3. Sparse attention promises theoretical speedup

Instead of computing full attention, sparse attention selectively processes only the important tokens based on sparsity patterns, leading to significant computational savings. We

---

training data consists of slow-motion videos, making the temporal head's focus on tokens with the same spatial position in several frames adequate to maintain temporal consistency.

**Algorithm 1** Online Profiling Strategy

```
# Q, K, V, O: [B, H, S, D] – query, key, value, output
# S:            – Total Token Number E.g., 18k
# t:            – Sampled Token Number. E.g., 32

# Sample the Indices
indices = sample_indices(S, t) # (t,)
Q_i = Q[:, :, indices, :]

# Get the attention masks
mask_spatial = gen_spatial_mask()[:, :, indices, :]
mask_temporal = gen_temporal_mask()[:, :, indices, :]

# Compute sampled attention score
# Shape: [B, H, t, D]
O_full = mask_attention(Q_i, K, V, None)
O_spatial = mask_attention(Q_i, K, V, mask_spatial)
O_temporal = mask_attention(Q_i, K, V, mask_temporal)

# Calculate MSE and get best mask
# Shape: [B, H]
MSE_s = (O_full - O_spatial).norm().mean(dim=(2,3))
MSE_t = (O_full - O_temporal).norm().mean(dim=(2,3))
best_mask_config = (MSE_s < MSE_t)
```

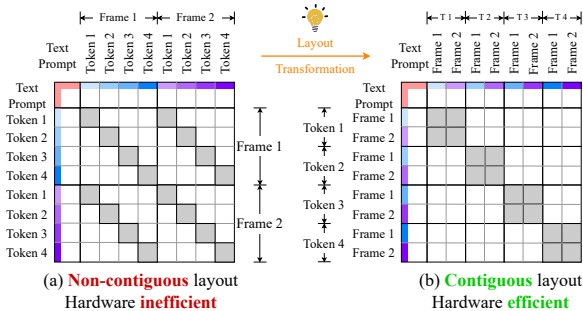

*Figure 5.* Visualization of hardware-efficient layout transformation. (a) Non-contiguous sparsity layout of temporal head, which is hardware inefficient due to the contiguous layout required by hardware accelerators. (b) Contiguous layout generated by transposing the token-major tensor into a frame-major one, which can be efficiently handled by block sparse attention.

analyze the theoretical computation saving below.

Given a model configuration of hidden dimension $H$, number of tokens per frame $L$, and number of total frames $N$, the total computation (FLOPS) for each full attention is $2 \cdot 2 \cdot (LN)^2 \cdot H = 4L^2N^2H$. For spatial head, assuming each head only attends to nearby $c_s$ frames, the computation is reduced to $(2 \cdot 2 \cdot L^2H) \cdot c_sN$, resulting in a sparsity of $\frac{c_s}{N}$. For temporal head, assuming each token only attends $c_t$ tokens across all the frames, the computation is $(2 \cdot 2 \cdot N^2H) \cdot c_tL$, with a sparsity of $\frac{c_t}{L}$. Since both $c_s$ and $c_t$ are typically much smaller compared to $N$ and $L$ respectively, the sparsity can easily achieve 30%. E.g., the aforementioned CogVideoX-v1.5-T2V achieves a sparsity of 31% for both spatial and temporal head while maintaining an average of 29.99 PSNR.

Despite the theoretical speedup, the temporal head can not be directly translated into real speedup since the pattern is hardware-inefficient. We will discuss our practical solution in Sec. 4.2 and prove it can achieve theoretical speedup in Sec. 5.3. Note that we do not include the text prompts and first frame in the theoretical calculation for simplicity, as they are constant and small compared to the remaining part.

# 4. Methodology

In this section, we introduce SVG, a training-free framework designed to exploit the sparse patterns of 3D full attention while addressing practical deployment challenges through careful design. To identify sparse patterns, SVG employs an online profiling strategy (Sec 4.1). To effectively utilize sparsity, SVG introduces a hardware-efficient layout transformation, which enables real-world hardware acceleration (Sec 4.2). Additionally, by integrating techniques such as customized kernels and quantization (Sec 4.3), SVG significantly accelerates video generation without compromising

generation quality.

## 4.1. Online profiling strategy for sparsity identification

As discussed in Sec 3.1, all attention heads can be classified and sparsified into spatial head and temporal head. However, we find that such sparse patterns can be *highly dynamic* across different denoising steps and input data. E.g., a certain head can be a spatial head for one prompt while being a temporal head given another. This dynamic nature necessitates an efficient online sparsity identification method, which classifies attention heads on the fly without extra overhead.

To this end, SVG proposes an *online profiling strategy*. Instead of computing the entire full attention to identify sparse attention, SVG only samples a subset of input rows ($x\%$) and calculates results with both the spatial and temporal sparsity patterns. By choosing the one with the lower MSE compared to full attention, SVG can efficiently approximate the oracle identification method discussed in Sec 3.2. We detail the profiling process in Algorithm 1.

To demonstrate the effectiveness of the proposed method, we conduct a sensitivity test on profiling ratio $x$ with CogVideoX-v1.5-I2V. As shown in Table 3, profiling only 1% can achieve up to 31.1 PSNR, with only 3% runtime overhead compared to full attention.

## 4.2. Hardware-efficient layout transformation

Despite the high sparsity in attention computation, speedups are limited without a hardware-efficient sparsity layout (Ye et al., 2023; Zheng et al., 2023). For instance, NVIDIA's Tensor Core, a matrix-matrix multiplication accelerator, requires at least 16 contiguous elements along each dimension to use. However, temporal head exhibits a sparse layout of non-contiguous elements with a stride of $L$ (i.e.,

*Table 1.* Quality and efficiency benchmarking results of SVG and other baselines.

| Type | Method | Quality | | | | | Efficiency | | |
|---|---|---|---|---|---|---|---|---|---|
| | | PSNR ↑ | SSIM ↑ | LPIPS ↓ | ImageQual ↑ | SubConsist ↑ | FLOPS ↓ | Latency ↓ | Speedup ↑ |
| **I2V** | CogVideoX-v1.5 (720p, 10s, 80 frames) | - | - | - | 70.09% | 95.37% | 147.87 PFLOPs | 528s | 1x |
| | DiTFastAttn (Spatial-only) | 24.591 | 0.836 | 0.167 | 70.44% | 95.29% | 78.86 PFLOPs | 338s | 1.56x |
| | Temporal-only | 23.839 | 0.844 | 0.157 | 70.37% | 95.13% | 70.27 PFLOPs | 327s | 1.61x |
| | MInference | 22.489 | 0.743 | 0.264 | 58.85% | 87.38% | 84.89 PFLOPs | 357s | 1.48x |
| | PAB | 23.234 | 0.842 | 0.145 | 69.18% | 95.42% | 105.88 PFLOPs | 374s | 1.41x |
| | Ours | **28.165** | **0.915** | **0.104** | 70.41% | 95.29% | 74.57 PFLOPs | 237s | **2.23x** |
| **T2V** | CogVideoX-v1.5 (720p, 10s, 80 frames) | - | - | - | 62.42% | 98.66% | 147.87 PFLOPs | 528s | 1x |
| | DiTFastAttn (Spatial-only) | 23.202 | 0.741 | 0.256 | 62.22% | 96.95% | 78.86 PFLOPs | 338s | 1.56x |
| | Temporal-only | 23.804 | 0.811 | 0.198 | 62.12% | 98.53% | 70.27 PFLOPs | 327s | 1.61x |
| | MInference | 22.451 | 0.691 | 0.304 | 54.87% | 91.52% | 84.89 PFLOPs | 357s | 1.48x |
| | PAB | 22.486 | 0.740 | 0.234 | 57.32% | 98.76% | 105.88 PFLOPs | 374s | 1.41x |
| | Ours | **29.989** | **0.910** | **0.112** | 63.01% | 98.67% | 74.57 PFLOPs | 232s | **2.28x** |
| **T2V** | HunyuanVideo (720p, 5.33s, 128 frames) | - | - | - | 66.11% | 93.69% | 612.37 PFLOPs | 2253s | 1x |
| | DiTFastAttn (Spatial-only) | 21.416 | 0.646 | 0.331 | 67.33% | 90.10% | 260.48 PFLOPs | 1238s | 1.82x |
| | Temporal-only | 25.851 | 0.857 | 0.175 | 62.12% | 98.53% | 259.10 PFLOPs | 1231s | 1.83x |
| | MInference | 23.157 | 0.823 | 0.163 | 63.96% | 91.12% | 293.87 PFLOPs | 1417s | 1.59x |
| | Ours | **29.546** | **0.907** | **0.127** | 65.90% | 93.51% | 259.79 PFLOPs | 1171s | 1.92x |
| | Ours + FP8 | **29.452** | **0.906** | **0.128** | 65.70% | 93.51% | 259.79 PFLOPs | 968s | **2.33x** |

number of tokens per frame). This sparsity pattern prevents effective utilization of Tensor Core, thereby constraining overall efficiency.

To tackle this, SVG introduces a *layout transformation* strategy that transforms the sparsity layout of temporal head into a hardware-efficient one. As illustrated in Figure 5, this strategy transposes a token-major tensor into a frame-major one, which makes the tokens across different frames into a contiguous layout. Such transformation maintains a mathematically equivalent output as attention computation is associative (Dao et al., 2022; 2019). We ablate the effectiveness of the proposed method in Sec 5.3.

### 4.3. Other optimizations

**Efficient kernel customization.** We notice that current implementations of QK-norm and RoPE suffer from performance issues, due to limited parallelism on small head dimensions (e.g., 64 in CogVideoX-v1.5). Therefore, we customize those operations with CUDA by a sub-warp reduction implementation, providing up to $5\times$ speedup compared to torch implementation (see Table 2). We also use Triton to implement fused online profiling strategy and layout transformation kernels, followed by a block sparse attention kernel with FlashInfer (Ye et al., 2025).

**Quantization.** We further incorporate **FP8 quantization** into sparse attention (Zhang et al., 2025b; 2024; Zhao et al., 2024c), which further boosts up to $1.3\times$ throughput with minimal accuracy drop as shown in Table 1. We also customize an attention kernel that supports both FP8 quantization and block sparse computation.

## 5. Experiment

### 5.1. Setup

**Models.** We evaluate SVG on open-sourced state-of-the-art video generation models including CogVideoX-v1.5-I2V, CogVideoX-v1.5-T2V, and HunyuanVideo-T2V to generate 720p resolution videos. After 3D VAE, CogVideoX-v1.5 consumes 11 frames with 4080 tokens per frame in 3D full attention, while HunyuanVideo works on 33 frames with 3600 tokens per frame.

**Metrics.** We assess the quality of the generated videos using the following metrics. We use Peak Signal-to-Noise Ratio (PSNR), Learned Perceptual Image Patch Similarity (LPIPS) (Zhang et al., 2018), Structural Similarity Index Measure (SSIM) to evaluate the generated video's similarity, and use VBench Score (Huang et al., 2023) to evaluate the video quality, following common practices in community (Horé & Ziou, 2010; Zhao et al., 2024b; Li* et al., 2025; Li et al., 2024). Specifically, we report the imaging quality and subject consistency metrics, represented by ImageQaul and SubConsist in our table.

**Datasets.** For CogVideoX-v1.5, we generate video using the VBench dataset after prompt optimization, as suggested by CogVideoX (Yang et al., 2024c). For HunyuanVideo, we benchmark our method using the prompt in Penguin Video Benchmark released by HunyuanVideo (Kong et al., 2024).

**Baselines.** We compare SVG against sparse attention algorithms DiTFastAttn (Yuan et al., 2024) and MInference (Jiang et al., 2024). As DiTFastAttn can be considered as spatial head only algorithm, we also manually implement a temporal head only baseline named *Temporal-only attention*. We also include a cache-based DiT acceleration

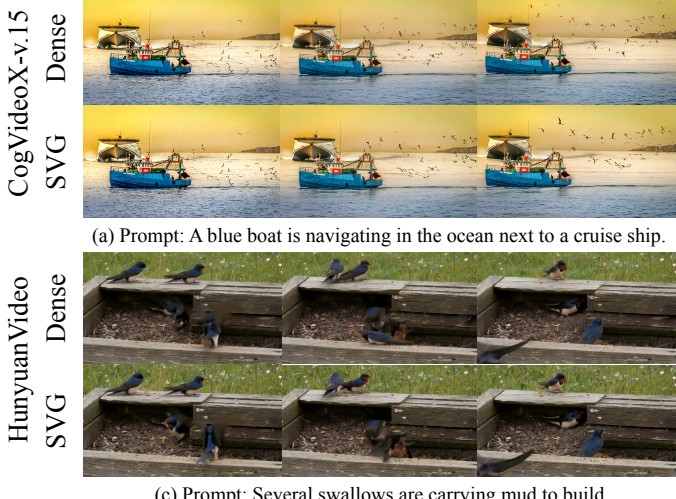

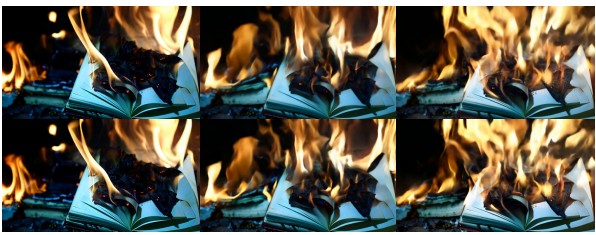

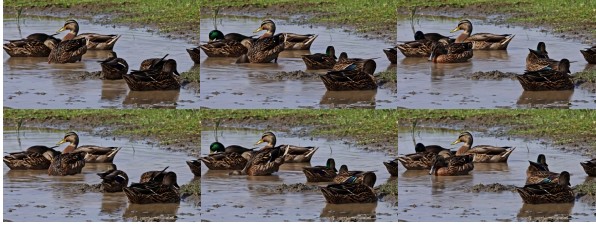

(a) Prompt: A blue boat is navigating in the ocean next to a cruise ship.

(b) Prompt: A book on fire with flames coming out of it.

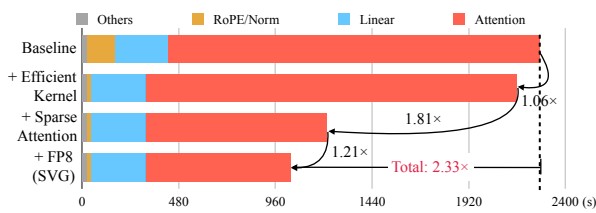

(c) Prompt: Several swallows are carrying mud to build nests under the eaves, low angle shot.

(d) Prompt: Several ducks are lying in the mud pit, occasionally preening their feathers leisurely.

*Figure 6.* Examples of generated videos by SVG and original implementation on CogVideoX-v1.5-I2V and HunyuanVideo-T2V. We showcase four different scenarios: (a) minor scene changes, (b) significant scene changes, (c) rare object interactions, and (d) frequent object interactions. SVG produces videos highly consistent with the originals in all cases, maintaining high visual quality.

*Figure 7.* The breakdown of end-to-end runtime of HunyuanVideo when generating a 5.3s, 720p video. SVG effectively reduces the end-to-end inference time from 2253 seconds to 968 seconds through system-algorithm co-design. Each design point contributes to a considerable improvement, with a total $2.33\times$ speedup.

algorithm PAB (Zhao et al., 2024b) as a baseline.

**Parameters.** For MInference and PAB, we use their official configurations. For SVG, we choose $c_s$ as 4 frames and $c_t$ as 1224 tokens for CogVideoX-v1.5, while $c_s$ as 10 frames and $c_t$ as 1200 tokens for HunyuanVideo. Such configurations lead to approximately 30% sparsity for both spatial head and temporal head, which is enough for lossless generation in general. We skip the first 25% denoising steps for all baselines as they are critical to generation quality, following previous works (Zhao et al., 2024b; Li et al., 2024; Lv et al., 2024; Liu et al., 2024a).

**Visualizations.** We present a comparison of the videos generated by Dense Attention and Sparse VideoGen in Appendix B. Additionally, real video samples are available on Google Drive and can be accessed here.

## 5.2. Quality evaluation

We evaluate the quality of generated videos by SVG compared to baselines and report the results in Table 1. Results demonstrate that SVG **consistently outperforms** all baseline methods in terms of PSNR, SSIM, and LPIPS while achieving **higher end-to-end speedup**.

Specifically, SVG achieves an average PSNR exceeding **29.55** on HunyuanVideo and **29.99** on CogVideoX-v1.5-T2V, highlighting its exceptional ability to maintain high fidelity and accurately reconstruct fine details. For a visual understanding of the video quality generated by SVG, please refer to Figure 6.

SVG maintains both **spatial and temporal consistency** by adaptively applying different sparse patterns, while all other baselines fail. E.g., since the mean-pooling block sparse cannot effectively select slash-wise temporal sparsity (see Figure 3), MInference fails to account for temporal dependencies, leading to a substantial PSNR drop. Besides, PAB skips computation of 3D full attention by reusing results from prior layers, which greatly hurts the quality.

Moreover, SVG is compatible with **FP8 attention quantization**, incurring only a 0.1 PSNR drop on HunyuanVideo. Such quantization greatly boosts the efficiency by $1.3\times$. Note that we do not apply FP8 attention quantization on CogVideoX-v1.5, as its head dimension of 64 limits the arithmetic intensity, offering no on-GPU speedups.

*Table 2.* Inference speedup of customized QK-norm and RoPE compared to PyTorch implementation with different number of frames. We use the same configuration of CogVideoX-v1.5, i.e. 4080 tokens per frame, 96 attention heads.

| Frame Number | 8 | 9 | 10 | 11 |
|---|---|---|---|---|
| QK-norm | 7.44× | 7.45× | 7.46× | 7.47× |
| RoPE | 14.50× | 15.23× | 15.93× | 16.47× |

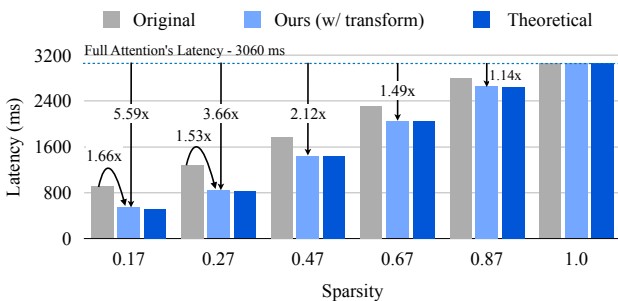

*Figure 8.* Latency comparison of different implementations of sparse attention. Our hardware-efficient layout transformation optimizes the sparsity pattern of temporal head for better contiguity, which is 1.7× faster than naive sparse attention (named original), approaching the theoretical speedup.

*Table 3.* Sensitivity test on online profiling strategy ratios. Profiling just 1% tokens achieves generation quality comparable to the oracle (100%) while introducing only negligible overhead.

| Ratios | PSNR ↑ | SSIM ↑ | LPIPS ↓ |
|---|---|---|---|
| **CogVideoX-v1.5-I2V (720p, 10s, 80 frames)** | | | |
| profiling 0.1% | 30.791 | 0.941 | 0.0799 |
| profiling 1% | 31.118 | 0.945 | 0.0757 |
| profiling 5% | 31.008 | 0.944 | 0.0764 |
| profiling 100% | 31.324 | 0.947 | 0.0744 |

### 5.3. Efficiency evaluation

To demonstrate the feasibility of SVG, we prototype the entire framework with dedicated CUDA kernels based on FlashAttention (Dao et al., 2022), FlashInfer (Ye et al., 2025), and Triton (Tillet et al., 2019). We first showcase the end-to-end speedup of SVG compared to baselines on an H100-80GB-HBM3 with CUDA 12.4. Besides, we also conduct a kernel-level efficiency evaluation. Note that all baselines adopt FlashAttention-2 (Dao et al., 2022).

**End-to-end speedup benchmark.** We incorporate the end-to-end efficiency metric including FLOPS, latency, and corresponding speedup into Table 1. SVG consistently outperforms all baselines by achieving an average speedup of 2.28× while maintaining the highest generation quality. We further provide a detailed breakdown of end-to-end inference time on HunyuanVideo in Figure 7 to analyze the speedup. Each design point described in Sec 4 contributes significantly to the speedup, with sparse attention delivering the most substantial improvement of 1.81×.

**Kernel-level efficiency benchmark.** We benchmark individual kernel performance including QK-norm, RoPE, and block sparse attention with unit tests in Table 2. Our customized QK-norm and RoPE achieve consistently better throughput across all frame numbers, with an average speedup of 7.4× and 15.5×, respectively. For the sparse attention kernel, we compare the latency of our customized kernel with the theoretical speedup across different sparsity. As shown in Figure 8, our kernel achieves theoretical speedup, enabling practical benefit from sparse attention.

### 5.4. Sensitivity test

In this section, we conduct a sensitivity analysis on the hyperparameter choices of SVG, including the online profiling strategy ratios (Sec 4.1) and the sparsity ratios $c_s$ and $c_t$ (Sec 4.2). Our goal is to demonstrate the robustness of SVG across various efficiency-accuracy trade-offs.

**Online profiling strategy ratios.** We evaluate the effectiveness of online profiling strategy with different profiling ratios on CogVideoX-v1.5 using a random subset of VBench in Table 3. In our experiments, we choose to profile only 1% of the input rows, which offers a comparable generation quality comparable to the oracle profile (100% profiled) with negligible overhead.

**Generation quality over different sparsity ratios.** As discussed in Sec 3.3, different sparsity ratio of the spatial head and temporal head can be set by choosing different $c_s$ and $c_t$, therefore reaching different trade-offs between efficiency and accuracy. We evaluate the LPIPS of HunyuanVideo over a random subset of VBench with different sparsity ratios. As shown in Table 4, SVG consistently achieves decent generation quality across various sparsity ratios. E.g., even with a sparsity of 13%, SVG still achieves 0.154 LPIPS. We leave the adaptive sparsity control for future work.

### 5.5. Ablation study

We conduct the ablation study to evaluate the effectiveness of the proposed hardware-efficient layout transformation (as discussed in Sec 4.2). Specifically, we profile the latency of the sparse attention kernel with and without the transformation under the HunyuanVideo configuration. As shown in Figure 8, the sparse attention with layout transformation closely approaches the theoretical speedup, whereas the original implementation without layout transformation falls short. For example, at a sparsity level of 10%, our method achieves an additional 1.7× speedup compared to

*Table 4.* Video quality of HunyuanVideo on a subset of VBench when varying sparsity ratios. LPIPS decreases as the sparse ratio increases, achieving trade-offs between efficiency and accuracy.

| Sparsity↓ | 0.13 | 0.18 | 0.35 | 0.43 | 0.52 |
|---|---|---|---|---|---|
| LPIPS↓ | 0.154 | 0.135 | 0.141 | 0.129 | 0.116 |

the original approach, achieving a $3.63\times$ improvement.

## 6. Conclusion

We accelerate video diffusion transformers by exploiting sparse attention. We reveal that attention heads have inherent sparsity patterns and we classify them into spatial head and temporal head. We proposed Sparse VideoGen (SVG), a training-free method to utilize these sparsity patterns for end-to-end efficiency boosts, including an efficient online profiling algorithm and an efficient inference system. On representative open video diffusion transformers (CogVideoX-v1.5 and HunyuanVideo), SVG demonstrates prominent end-to-end speedup without losing quality.

## Impact Statement

This paper presents work that aims to advance the field of Machine Learning. There are many potential societal consequences of our work, none of which we feel must be specifically highlighted here.

## Acknowledgment

We thank Guangxuan Xiao and Zihao Ye for the visualizations and kernel designs.

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

# A. A full version of related work

## A.1. Efficient Diffusion Models

Diffusion Models function primarily as denoising models that are trained to estimate the gradient of the data distribution (Song & Ermon, 2019; Zhang et al., 2023a). Although these models are capable of generating samples with high quality and diversity, they are known as inefficient. To enhance the efficiency of diffusion models, researchers often focus on three primary approaches: (1) decreasing the number of denoising steps, (2) reducing the model size, and (3) optimizing system implementation for greater efficiency.

**Decreasing the denoising steps.** The main diffusion models rely on stochastic differential equations (SDEs) that learn to estimate the gradient of the data distribution through Langevin dynamics (Ho et al., 2020; Meng et al., 2022). Consequently, these models generally require numerous sampling steps (, *e.g.,* 1,000). To improve sample efficiency, DDIM (Song et al., 2020) approximates SDE-based diffusion models within an ordinary differential equation (ODE) framework. Expanding on this concept, DPM (Lu et al., 2022a), DPM++ (Lu et al., 2022b), and Rectified Flows (Liu et al., 2022; 2024c) enhance ODE paths and solvers to further reduce the number of denoising steps. Furthermore, Consistency Models (Song et al., 2023; Luo et al., 2023) integrate the ODE solver into training using a consistency loss, allowing diffusion models to replicate several denoising operations with fewer iterations. In addition, approaches grounded in distillation (Yin et al., 2024a;b) represent another pivotal strategy. This involves employing a simplified, few-step denoising model to distill a more complex, multi-step denoising model, thereby improving overall efficiency.

Nevertheless, all these approaches necessitate either re-training or fine-tuning the complete models on image or video datasets. For video generation models, this is largely impractical due to the significant computational expense involved, which is prohibitive for the majority of users. In this work, our primary focus is on a method to enhance generation speed that requires no additional training.

**Diffusion Model Compreesion** A common approach to enhancing the efficiency of diffusion models involves compressing their weights through quantization. Q-Diffusion (Li et al., 2023) introduced a W8A8 strategy, implementing quantization in these models. Building on this foundation, ViDiT-Q (Zhao et al., 2024a) proposed a timestep-aware dynamic quantization method that effectively reduces the bit-width to W4A8. Furthermore, SVDQuant (Li* et al., 2025) introduced a cost-effective branch designed to address outlier problems in both activations and weights, thus positioning W4A4 as a feasible solution for diffusion models. SageAttention (Zhang et al., 2025b) advanced the field by quantizing the attention module to INT8 precision via a smoothing technique. SageAttention V2 (Zhang et al., 2024; 2025d) and V3 (Zhang et al., 2025a) extended these efforts by pushing the precision boundaries to INT4 hybridized with FP8 or even full FP4. Another common approach is to design efficient diffusion model architectures (Xie et al., 2024; Cai et al., 2024; Chen et al., 2025) and high-compression autoencoders (Chen et al., 2024a) to boost efficiency. Our Sparse VideoGen is orthogonal to these techniques and can utilize them as supplementary methods to enhance efficiency.

**Efficient System Implementation** In addition to enhancing the efficiency of diffusion models by either retraining the model to decrease the number of denoising steps or compressing the model size, efficiency improvements can also be achieved at the system level. For instance, strategies such as dynamic batching are employed in StreamDiffusion (Kodaira et al., 2023) and StreamV2V (Liang et al., 2024) to effectively manage streaming inputs in diffusion models, thereby achieving substantial throughput enhancements. Other approaches include: DeepCache (Ma et al., 2024), which leverages feature caching to modify the UNet Diffusion; $\Delta - DiT$ (Chen et al., 2024b), which implements this mechanism by caching residuals between attention layers in DiT to circumvent redundant computations; and PAB (Zhao et al., 2024b), which caches and broadcasts intermediary features at distinct timestep intervals. FasterCache (Lv et al., 2024) identifies significant redundancy in CFG and enhances the reuse of both conditional and unconditional outputs. Meanwhile, TeaCache (Liu et al., 2024a) recognizes that the similarity in model inputs can be used to forecast output similarity, suggesting an improved machine strategy to amplify speed gains.

Despite these advanced methodologies, they often result in the generated output diverging significantly from the original, as indicated by a PSNR falling below 22. In contrast, our method consistently achieves a PSNR exceeding 30, thus ensuring substantially superior output quality compared to these previously mentioned strategies.

### A.2. Efficient Attention

**Sparse Attention in LLM**   Recent studies on sparse attention in language models have identified patterns that reduce computational costs by targeting specific token subsets. StreamingLLM (Xiao et al., 2023) and LM-Infinite (Han et al., 2023) reveal concentration on initial and local tokens, highlighting temporal locality in decoding. H2O (Zhang et al., 2023b) and Scissorhands (Liu et al., 2024d) note attention focuses mainly on a few dominant tokens. TidalDecode (Yang et al., 2024a) shows cross-layer attention pattern correlations, aiding in attention sparsity. DuoAttention (Xiao et al., 2024a) and MInference (Jiang et al., 2024) find distinct sparse patterns among attention heads, with varying focus on key tokens and context. MMInference  (Li et al., 2025) speedup the vision language models through modality-aware permutation. SpargeAttention (Zhang et al., 2025c) and XAttention (Xu et al., 2025) propose general sparsity identification algorithms that can be applied to all forms of models. Despite their success in LLMs or VLMs, these mechanisms are constrained to token-level sparsity and miss the redundancy unique to video data.

**Linear and Low-bit Attention**   Significant advancements have been achieved in enhancing attention efficiency, notably through linear attention (Cai et al., 2023; Xie et al., 2024) and low-bit attention techniques (Zhang et al., 2025b; 2024). Linear attention models, including Linformer (Wang et al., 2020), Performer (Choromanski et al., 2020), MetaFormer (Yu et al., 2022), and LinearAttention (Katharopoulos et al., 2020), reduce the quadratic complexity of traditional attention to linear. Low-bit attention approaches decrease computational demands by utilizing lower precision, with SageAttention (Zhang et al., 2025b) employing INT8 precision to enhance efficiency without notable performance loss.

Sparse VideoGen, as a **sparse attention** method, is **orthogonal** to both linear attention and low-bit attention techniques. Moreover, it can be integrated with low-bit attention methods, such as FP8 attention, to further enhance computational efficiency.

## B. Visualization of the generated videos

We provide visualization comparison between Dense Attention and Sparse VideoGen on HunyuanVideo and Wan 2.1. We conduct both Text-to-Video generation and Image-to-Video generation under 720p resolution. Results demonstrate that Sparse VideoGen can preserve high pixel-level fidelity, achieving similar generation quality compared with the dense attention.

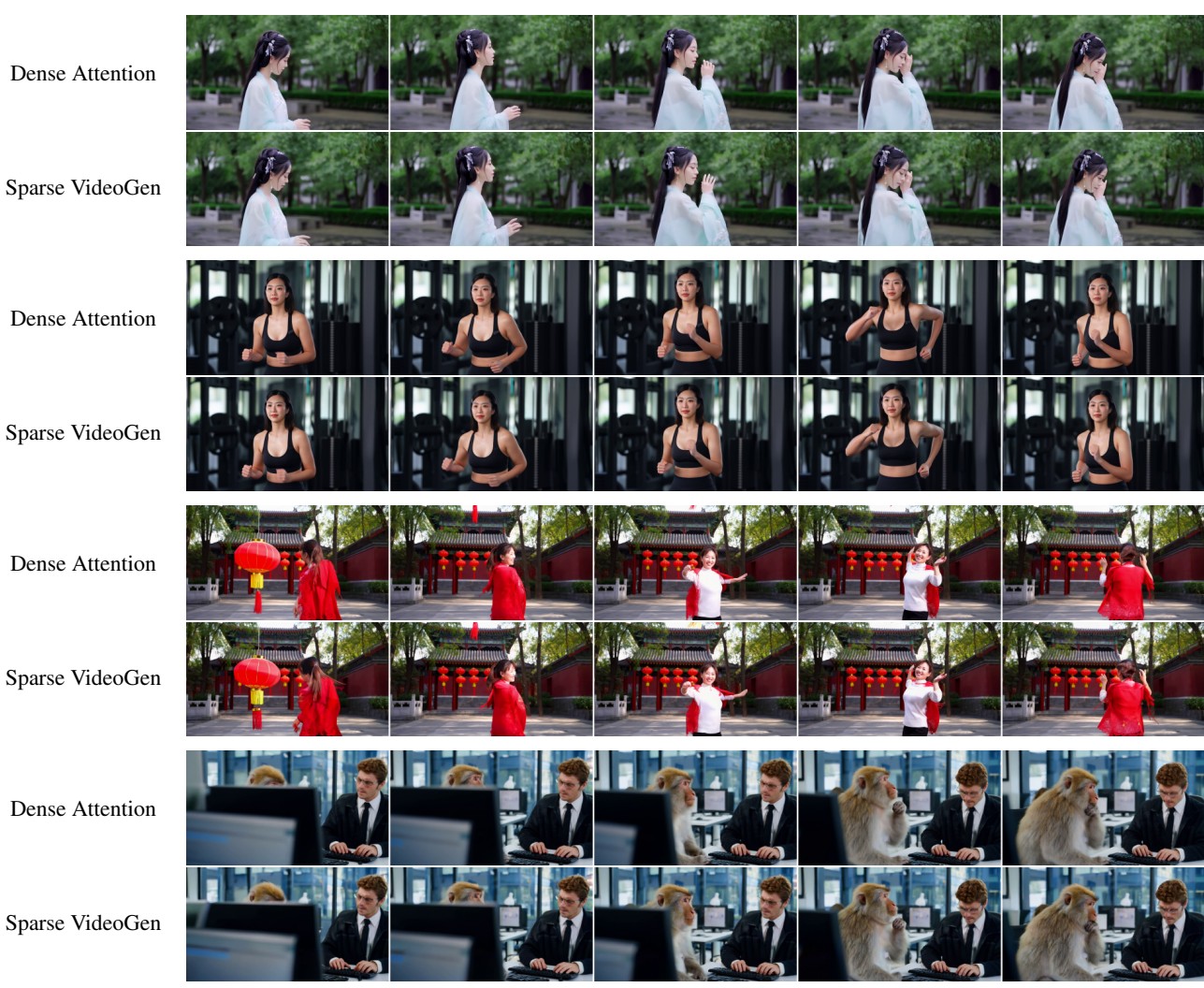

*Figure 9.* Comparion of Dense Attention and Sparse VideoGen on HunyuanVideo Text-to-Video generation.

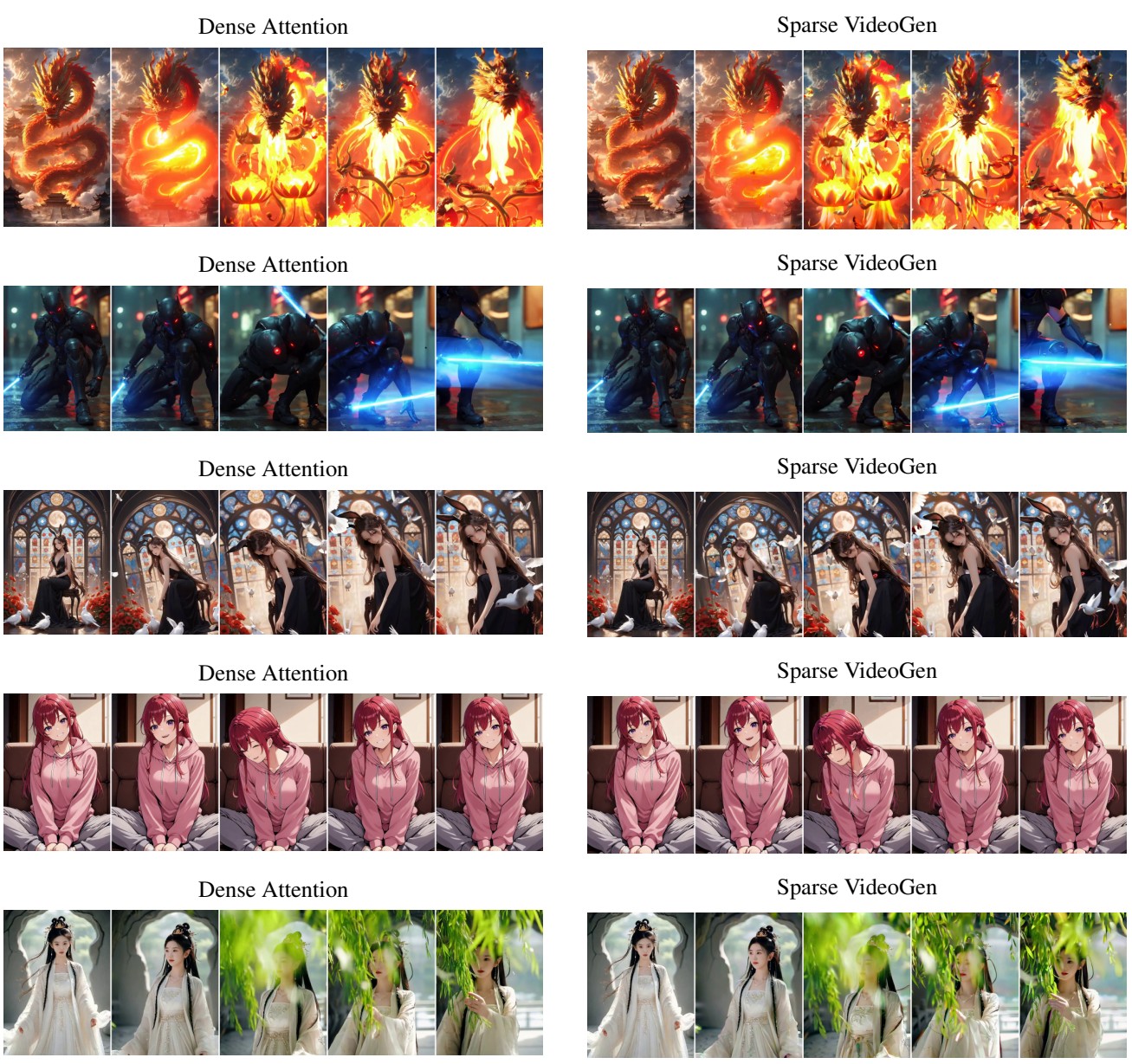

*Figure 10.* Comparison of Dense Attention and Sparse VideoGen on Wan 2.1 Image-to-Video generation.

# C. Theoretical Analysis of Sparse Video Attention

Here we provide the theoretical analyses of SVG. Consider a video with $N$ frames, each containing $F$ tokens. Every token can be encoded by an indices pair $(i, j)$, where $0 \leq i < N, 0 \leq j < F$. In the attention map we will flatten these two-dimensional indices into a 1D vector. Corresponding formula is $x = i \cdot F + j$.

## Spatial Head

For spatial head, let $a_1 > 0$ denote the threshold for spatial closeness between tokens. The spatial attention mask is defined as:

$$f_s((i_1, j_1), (i_2, j_2)) = \begin{cases} 1 & \text{if } |i_1 - i_2| < a_1, \\ 0 & \text{otherwise} \end{cases}$$

For flattened indices $x_1 = i_1 \cdot F + j_1$ and $x_2 = i_2 \cdot F + j_2$,

$$f_s(x_1, x_2) = 1 \Leftrightarrow \left\lfloor \frac{x_1}{F} \right\rfloor - \left\lfloor \frac{x_2}{F} \right\rfloor < a_1$$

The resulting attention map takes the form of block-banded structures: the attention mask will be on the main diagonal and also on neighboring $\pm(a_1 - 1)$ diagonals.

## Temporal Head

For temporal head, let $a_2 > 0$ denote the threshold for temporal closeness between tokens. The temporal attention mask is defined as:

$$f_t((i_1, j_1), (i_2, j_2)) = \begin{cases} 1 & \text{if } |j_1 - j_2| < a_2, \\ 0 & \text{otherwise} \end{cases}$$

For flattened indices $x_1 = i_1 \cdot F + j_1$ and $x_2 = i_2 \cdot F + j_2$,

$$f_t(x_1, x_2) = 1 \Leftrightarrow |(x_1 \bmod F) - (x_2 \bmod F)| < a_2 \Leftrightarrow \exists k, 0 \leq k < 2N - 1, |(x_1 - x_2) - kF| < a_2$$

The resulting attention map forms $2N - 1$ slanted diagonals that align along constant column index differences. These diagonals, often referred to as "slashes," correspond to the token positions sharing similar temporal locations across frames. The corresponding points in the attention matrix yield a "slash-wise" pattern with width $a_2$.

