# OpenReview forum: "Sparse Video-Gen: Accelerating Video Diffusion Transformers with Spatial-Temporal Sparsity"
_ICML.cc/2025/Conference — ICML 2025 poster_

### Official Review · Reviewer_7rqP · 2025-03-05

**Overall Recommendation:** 3

**Summary:**

The paper presents Sparse Video-Gen (SVG), a training-free framework aimed at accelerating the inference of video diffusion models. Diffusion Transformers (DiTs) are powerful for video generation but suffer from high computational costs due to the quadratic complexity of 3D full attention. The authors reveal two distinct sparse attention patterns in DiTs: Spatial Head and Temporal Head. SVG uses an online profiling strategy to identify these patterns for sparse attention and a novel hardware - efficient tensor layout transformation, along with customized kernel implementations. It achieves significant speedups (up to 2.28x and 2.33x on CogVideoX and HunyuanVideo respectively) while maintaining high-quality video generation.

**Claims And Evidence:**

Claims: The main claims are that SVG can accelerate video diffusion models, identify distinct sparse attention patterns, and achieve high-quality video generation at a faster speed.

**Essential References Not Discussed:**

No essential references were identified as missing from the paper. The authors have covered a wide range of relevant works in the field of video generation, efficient attention, and diffusion models.

**Experimental Designs Or Analyses:**

The analyses of the experimental results are detailed.

**Methods And Evaluation Criteria:**

Yes

**Other Comments Or Suggestions:**

No

**Other Strengths And Weaknesses:**

### **Strengths**

1. **Sufficient Model Evaluation**: SVG is evaluated on representative open-sourced video generation models, including CogVideoX and HunyuanVideo.

### **Weaknesses**

1. **Limited Technical Depth**: The paper appears to be mainly an empirical exploration. it lacks in-depth technical analysis beyond the proposed online profiling strategy and layout transformation. For instance, the theoretical analysis of the sparsity patterns mainly focuses on the computational savings and does not delve deeply into the underlying mathematical principles or theoretical insights that govern these sparse patterns in the context of video diffusion models. This makes the technical contribution seem rather shallow in the ICML context.
2. **Superficial Understanding of Sparsity**: In this paper, while they identify two types of attention heads based on sparse patterns (Spatial Head and Temporal Head), they do not provide a comprehensive exploration of why these patterns emerge at a fundamental level. There is no in-depth discussion on how the nature of video data, such as temporal continuity and spatial structure, precisely relates to these sparse patterns. Without a more thorough understanding of the root causes of sparsity, it is difficult to fully assess the generality and scalability of the proposed method.
3. **No video sample.** It is weird that a video generation paper do not have videos in the supplementary.


Overall, I hold a positive perspective on this paper. Addressing the aforementioned issues will significantly enhance the paper’s quality.

**Questions For Authors:**

No

**Relation To Broader Scientific Literature:**

The paper is well-related to the broader scientific literature. It builds on previous work in efficient diffusion models, efficient attention and sparse attention. The authors discuss how their work is orthogonal to existing techniques such as decreasing denoising steps, diffusion model compression, and linear/low - bit attention, and can be combined with them for further gains. They also contrast their approach with existing sparse attention methods in LLMs, highlighting that those methods do not leverage the inherent redundancy of video data.

**Theoretical Claims:**

This paper do not contain theoretical claims.

---

> ### Author Rebuttal · Authors · 2025-04-01
>
> We sincerely appreciate your insightful feedback and the opportunity to discuss our work further. Your comments have been invaluable in refining our approach and clarifications.
>
> > Weakness 1: In-depth technical analysis and underlying mathematical principles or theoretical insights that govern these sparse patterns in the context of video diffusion models.
>
> Answer:  Here we provide the theoretical analyses of SVG.
> Consider a video with N frames, each containing F tokens. Every token can be encoded by an indices pair $(i, j)$, where $0 \leq i < N, 0 \leq j < F$. In the attention map we will flatten these two-dimensional indices into a 1D vector. Corresponding formula is $x = i \cdot F + j$.
>
> **Spatial Head**:
>
> For spatial head, let $a_1 > 0$ denote the threshold for spatial closeness between tokens. The spatial attention mask is defined as:
>
> $f_{s}\left( (i_1, j_1), (i_2, j_2) \right) = 1 \text{ if } |i_1 - i_2| < a_1,  \text{ otherwise } 0$
>
> For flattened indices $x_1 = i_1 \cdot F + j_1$ and $x_2 = i_2 \cdot F + j_2$,
> $f_{s}(x_1, x_2) = 1$ is equivalent to:
> $| \lfloor \frac{x_1}{F} \rfloor - \lfloor \frac{x_2}{F} \rfloor | < a_1$
>
> The resulting attention map takes the form of block-banded structures: the attention mask will be on the main diagonal and also on neighboring $\pm(a_1 - 1)$ diagonals.
>
> **Temporal Head**
>
> For temporal head, let $a_2 > 0$ denote the threshold for temporal closeness between tokens. The temporal attention mask is defined as:
>
> $$
> f_{t}\left( (i_1, j_1), (i_2, j_2) \right) =
> 1 \text{ if } |j_1 - j_2| < a_2,  \text{ otherwise } 0
> $$
>
> For flattened indices $x_1 = i_1 \cdot F + j_1$ and $x_2 = i_2 \cdot F + j_2$,
> $f_{t}(x_1, x_2) = 1 \Leftrightarrow
> | (x_1 \bmod F) - (x_2 \bmod F) | < a_2 \Leftrightarrow
> | x_1 - F \cdot \lfloor \frac{x_1}{F} \rfloor - \left(x_2 - F \cdot \lfloor \frac{x_2}{F} \right) | < a_2 \Leftrightarrow
> \exists k, 0 <= k < 2N - 1, |(x_1 - x_2) - k F| < a_2$
>
> The resulting attention map forms $2N - 1$ slanted diagonals that align along constant column index differences. These diagonals, often referred to as “slashes,” correspond to the token positions sharing similar temporal locations across frames. The corresponding points in the attention matrix yield a “slash-wise” pattern with width $a_2$.
>
> ---
>
>
> In addition, we would like to emphasize that our work also provides other technical depth beyond the proposed online profiling and layout transformation. Our contributions comprise four main aspects.
>
> (1) We identify the spatial and temporal sparse patterns, which unveil the potential for efficient acceleration.
>
> (2) We propose algorithmic innovations, including online profiling strategy and layout transformation that exploit these patterns.
>
> (3) We co-design highly optimized kernels with our algorithm to translate theoretical computation saving to on-hardware speedup.
>
> (4) We present thorough evaluations that SVG can achieve an end-to-end speedup of up to 2.33x while maintaining high visual quality with a PSNR of up to 29.
>
> > Weakness 2: Comprehensive exploration of why these patterns emerge at a fundamental level.
>
> Answer: In essence, these patterns arise because the pixels in a video often exhibit **high similarity** both within a single frame (spatially) and across consecutive frames (temporally). As similar patches in a video share nearly identical embeddings in the Q/K/V space, the self-attention mechanism naturally assigns high attention scores to these similar patches.
>
> Spatial Head: Within a single frame, neighboring pixels (and patches) often share similar colors, brightness values, and gradual transitions. If a patch in one region of a frame has a high attention score to itself, it tends to exhibit high attention to other patches in the same frame that share similar visual patterns.
>
> Temporal Head: Consecutive frames in a video, especially in static or slowly changing scenes, exhibit high similarity for the same patch location across time. If a particular patch has a strong correlation with itself (i.e., high attention to its own embedding), it will also place high attention on patches in other frames that share similar visual features.
>
> Traditional video compression techniques have similarly leveraged spatial and temporal redundancy to reduce data size and complexity. For example, H264 (used in MP4) uses intra-prediction and discrete cosine transform (DCT) to reduce spatial redundancy and uses inter-prediction and recording only the difference (motion vectors and residuals) between consecutive frames to reduce temporal redundancy. These well-known strategies highlight that video data inherently contains a high degree of spatial and temporal redundancy—precisely the property our sparse videogen approach leverages.
>
> > Weakness 3: No video sample.
>
> Video samples showcasing the results of our method can be accessed at the following anonymous link:
> https://drive.google.com/drive/folders/1jhEpZ69bKfyZWmoy63iS3FhECNnX-AZU?usp=sharing

---

### Official Review · Reviewer_bnzg · 2025-03-12

**Overall Recommendation:** 4

**Summary:**

This paper proposes Sparse VideoGen for efficient (accelerated) video generation, a method which can be applied to existing video generative diffusion models which use diffusion transformers. The paper mainly targets video generative models which operate on the spatio-temporal latents. The key idea is to classify the inherent sparsity in the attention heads into either the spatial head or the temporal head based on their sparsity patterns, and to sparsity the attention operation based on the classified result. The authors notice that the sparsity patterns can be dynamic i.e., the same head can exhibit varying sparsity patterns depending on the prompt or timestep. To handle this, the authors propose an online profiling strategy to determine the inherent sparsity pattern on-the-fly. Combined with the proposed hardware-aware tensor layout transformation and customized kernel implementations of the QK-norm and rotary positional embedding (RoPE), SVG achieves SoTA speedup of existing (spatio-temporal) video generative models.

**Claims And Evidence:**

Yes, the claims made in the submission are supported by clear and convincing evidence e.g., lossless accuracy in 3.2 supported by PSNR value, quality and efficiency results in Table 1, and sensitivity test on online profiling strategy ideas in Table 3.

**Essential References Not Discussed:**

I could not find any significant essential reference which was not discussed.

**Experimental Designs Or Analyses:**

The experimental designs and analyses seem sound and valid.

**Methods And Evaluation Criteria:**

The proposed method and the evaluation criteria (videos generated from prompts provided in VBench, quality: SPNR, SSIM, LPIPs, VBench scores, and Efficiency: FLOPs, Latency, Speedup) make sense for the problem or application at hand.

**Other Comments Or Suggestions:**

No other comments or suggestions.

**Other Strengths And Weaknesses:**

Strengths \
S1. Strong performances in terms of efficiency and quality.  \
S2. Comprehensive experiments to validate the efficacy of each contribution made by SVG. \
S3. Well written and easy to follow.

Weaknesses
- I could not find any notable weaknesses from the paper.

**Questions For Authors:**

Once each attention head has been classified and sparsified to spatial / temporal heads, the overall pipeline may resemble video generative models which operate in the spatial domain i.e., pipeline consisting of interleaved spatial attention and temporal attention (e.g., SVD). How does the performance / efficiency compare to such models, which do not use 3D full attention? Such comparison would be valuable in understanding where SVG stands in the overall literature of video generative models.

**Relation To Broader Scientific Literature:**

As the authors have already mentioned, sparsity in transformers offer a great opportunity to reduce redundant computation e.g., in LLMs. However, the transformer architecture is likely to show largely varying sparsity patterns based on the task / field at hand e.g., video data has fundamentally different sparsity patterns. The findings in this work may be valuable for other fields where transformer networks are prominent e.g., MLLMs.

**Theoretical Claims:**

The paper does not propose proofs or theoretical claims that require correctness checks -- the claim that 'sparse attention achieves lossless accuracy' has been sufficiently validated.

---

> ### Author Rebuttal · Authors · 2025-04-01
>
> We thank the reviewer for acknowledging our contributions and the insightful comments. We respond to the questions below. We will revise the paper's final versions based on the reviewers' comments.
>
> > How does the performance / efficiency compare to models interleaving spatial attention and temporal attention (2D + 1D Attention)?
>
> Answer:
> Recent state-of-the-art models [1,2,3] have shifted toward using 3D full attention, which significantly improves performance at the cost of higher computational complexity. In contrast, although interleaved spatial and temporal attention [4,5,6] is computationally efficient, it suffers from **limited modeling capability**—especially in capturing complex spatiotemporal dependencies—leading to lower generation quality.
>
> Our method delivers video **generation quality on par with full 3D attention** models while operating at a **significantly lower computational cost**. Compared to 2D + 1D approaches, SVG achieves not only substantial improvements in visual fidelity but also stronger temporal consistency. In terms of efficiency, SVG sits between 2D + 1D and full 3D attention—achieving a compelling **balance of speed and quality**. For example, as shown in the table below, OpenSora-v1.2—a representative open-sourced model utilizing 2D+1D attention—achieves significantly lower VBench scores than Wan2.1, a 3D model of similar parameter size. By reducing attention redundancy, SVG effectively decreases attention FLOPs by 2.55x while preserving superior generation quality. Thus, SVG presents a practical and efficient alternative for high-quality video generation.
>
> | Model         | Attention Pattern | #Params | Attention FLOPS | VBench |
> |---------------|-------------------|---------|-----------------|--------|
> | OpenSora-v1.2 | 2D+1D             | 1.1B    | 8.4×10^12       | 77.59  |
> | Wan2.1        | 3D                | 1.3B    | 7.4×10^13       | 82.91  |
> | Wan2.1        | SVG               | 1.3B    | 2.9×10^13       | 82.91  |
>
> [1] Kong, Weijie, et al. "Hunyuanvideo: A systematic framework for large video generative models." arXiv preprint arXiv:2412.03603 (2024).
>
> [2] Wang, Ang, et al. "Wan: Open and Advanced Large-Scale Video Generative Models." arXiv preprint arXiv:2503.20314 (2025).
>
> [3] Yang, Zhuoyi, et al. "Cogvideox: Text-to-video diffusion models with an expert transformer." arXiv preprint arXiv:2408.06072 (2024).
>
> [4] Zheng, Zangwei, et al. "Open-sora: Democratizing efficient video production for all." arXiv preprint arXiv:2412.20404 (2024).
>
> [5] Lin, Bin, et al. "Open-sora plan: Open-source large video generation model." arXiv preprint arXiv:2412.00131 (2024).
>
> [6] Ma, Xin, et al. "Latte: Latent diffusion transformer for video generation." arXiv preprint arXiv:2401.03048 (2024).

---

> > ### Comment · Reviewer_bnzg · 2025-04-03
> >
> > Thank you very much for the response! The provided results are valuable in understanding that given similar number of parameters, leveraging SVG shows a high decrease in the attention FLOPs in comparison to using a full 3D attention, but still incurs higher FLOPs then the 2D+1D attention pattern -- which aligns with the response which mentions that "SVG sits between 2D + 1D and full 3D attention".
> >
> > I believe these results are valuable in understanding where SVG stands in the overall literature of video generative models, from the perspective of efficiency.
> >
> > Having read through other reviewers' comments and the corresponding author responses, I do not have further comments or concerns.

---

### Official Review · Reviewer_JiXV · 2025-03-14

**Overall Recommendation:** 3

**Summary:**

This paper addresses the critical challenge of computational inefficiency in Diffusion Transformers (DiTs) for video generation, caused by the quadratic complexity of 3D full attention operations. To tackle this issue, the authors propose Sparse VideoGen (SVG), a training-free inference optimization framework leveraging the inherent sparsity within 3D attention. Specifically, they reveal attention heads can be dynamically classified into two categories: Spatial Heads, dominated by spatially-related tokens within frames, and Temporal Heads, dominated by temporally-related tokens across frames. SVG employs an online profiling strategy to dynamically identify these sparse patterns during inference, coupled with a hardware-efficient tensor layout and customized kernel implementations.


###Post-rebuttal

Thank the authors for the rebuttal. I have no further questions and will keep my original score, leaning towards accepting the paper.

**Claims And Evidence:**

The claims presented in the submission are supported by experimental evidence.

**Essential References Not Discussed:**

I did not identify any essential related works that are missing from the current citations.

**Experimental Designs Or Analyses:**

Here are two key questions highlighting potential experimental concerns:

**Lack of Real Video Demonstrations**: The submission currently does not provide actual video demos, making it challenging to clearly evaluate the impact of sparse attention acceleration on generated video quality. It is better for authors to provide actual generated video examples (particularly corresponding to experiments shown in Figure 1 and Figure 6)? These demos would clarify how much video quality is affected by your acceleration method.

**Performance on Challenging Scenes**: How does the method perform in more challenging scenarios? Specifically, could the authors clearly demonstrate the performance on difficult cases, such as those involving complex spatial-temporal dynamics (e.g., rapid movements, human interactions such as group dancing, fighting, or dynamic sports scenes)? Providing video demonstrations of challenging scenes—such as a football match, basketball gameplay, rope skipping, fast-paced dancing, or multi-person interactions—would strengthen the claims regarding spatial and temporal consistency.

**Methods And Evaluation Criteria:**

The proposed methods and evaluation criteria are appropriate.

**Other Comments Or Suggestions:**

I have no additional comments or suggestions.

**Other Strengths And Weaknesses:**

**Additional Weakness**: Insufficient Clarity in Implementation Details. The manuscript currently lacks clarity regarding the implementation specifics of the proposed online profiling strategy. It is unclear exactly when this profiling occurs—does the profiling need to be computed at every diffusion step, or is it computed just once or intermittently? Moreover, does the profiling dynamically vary across different prompts, layers, or diffusion timesteps? Clarifying and validating this would strengthen the paper. Specifically, could the authors provide experimental evidence or analysis showing the stability or variability of profiling patterns under different conditions (e.g., across prompts, layers, or timesteps)?

**Questions For Authors:**

Please answer the questions in the experimental analyses and Implementation Details

**Relation To Broader Scientific Literature:**

The key contributions of this paper are closely related to the broader literature on efficient inference techniques for generative models, particularly diffusion-based video generation.

**Theoretical Claims:**

The paper does not contain explicit theoretical claims, formal proofs, or mathematical theorems that require verification. Therefore, there were no theoretical proofs to check.

---

> ### Author Rebuttal · Authors · 2025-04-01
>
> Thanks for your valuable comments and feedback. We respond to the questions below.
>
> > Experimental Designs Or Analyses 1: The submission currently does not provide actual video demos.
>
> Answer: As requested by the reviewer, we include actual video demonstrations at the following anonymous link:
> https://drive.google.com/drive/folders/1jhEpZ69bKfyZWmoy63iS3FhECNnX-AZU?usp=sharing. Our method achieves **nearly lossless** video quality.
>
> >  Experimental Designs Or Analyses 2: How does the method perform in more challenging scenarios? Such as those involving complex spatial-temporal dynamics.
>
> Answer: As suggested by the reviewer, we additionally include some representative video results of complex spatial-temporal dynamics in the following anonymous link: https://drive.google.com/drive/folders/1xscj4RcaOE-PeAXq6yiiUFqnTJqS5kaV?usp=sharing. Our method maintains **high video fidelity (PSNR=28.624)** under these challenging cases, such as scenes involving rapid motion and dynamic multi-person interactions.
>
> > Weakness 1: Insufficient clarity regarding the implementation specifics of the proposed online profiling strategy. When this profiling occurs, and does the profiling dynamically vary across different prompts, layers, or diffusion timesteps?
>
> Answer: Thank you for the valuable feedback. In our implementation, online profiling is performed at **every diffusion step** and **every layer** right before the 3D full-attention computation. This ensures that SVG dynamically adapts to the different prompts and spatial-temporal patterns at each step.
>
> We would like to emphasize that dynamic online profiling is essential to our method. We find that attention patterns exhibit **substantial variation**, especially across different steps and prompts. Static profiling patterns—whether fixed across layers, steps, or prompts—will lead to **performance degradation**, as demonstrated in Table 1, measured by PSNR, SSIM and LPIPS. The results are averaged over 10 examples. Our SVG uses the dynamic profiling strategy, consistently outperforming all static configurations.
>
> Table 1
> |                        | PSNR (↑) | SSIM (↑) | LPIPS (↓) |
> |------------------------|----------|----------|-----------|
> | Static Across Layers   | 22.365   | 0.846    | 0.187     |
> | Static Across Steps    | 28.007   | 0.866    | 0.131     |
> | Static Across Examples | 26.955   | 0.803    | 0.185     |
> | SVG (Ours)             | **29.062**   | **0.879**   | **0.122**     |
>
> Static profiling patterns lead to significant quality degradation due to inaccurate identification of attention patterns. To illustrate this, we analyze the variation in attention patterns across different layers, diffusion steps, and data examples (prompts) by computing cosine similarity. Specifically, we employ oracle profiling to identify attention patterns for each head across different configurations. For example, we calculate the variation across different layers by computing a tensor with shape [num_layers, num_heads]. Each element in this tensor is labeled either -1 (indicating a spatial head) or 1 (indicating a temporal head). By calculating the cosine similarity between columns of this tensor, we quantify the variation of attention patterns. As demonstrated in Table 2, attention patterns exhibit **substantial variation**, necessitating a dynamic online profiling method.
>
> Table 2
> |                   | Across Layers | Across Steps | Across prompts |
> |-------------------|---------------|--------------|----------------|
> | Cosine Similarity | 0.2597        | 0.7158       | 0.7238         |

---

### Official Review · Reviewer_usfF · 2025-03-15

**Overall Recommendation:** 3

**Summary:**

This paper proposes Sparse VideoGen, a training-free method to optimize and accelerate video diffusion DiTs through an online profiling strategy and hardware-friendly implementation.

**Claims And Evidence:**

Yes

**Essential References Not Discussed:**

NA

**Experimental Designs Or Analyses:**

Yes

**Methods And Evaluation Criteria:**

Yes

**Other Comments Or Suggestions:**

I am not familiar with this field.  However, I think the proposed training-free method is effective in accelerating the large video diffusion model. Therefore, I tend to give a weak accept.

**Other Strengths And Weaknesses:**

Strengths:

1. The observation that the attention layers in video diffusion can be divided into two types—pure spatial and pure temporal attention—is interesting and insightful.

2. The proposed online profiling strategy is well-defined, as the attention type in DiTs is highly dynamic depending on input data and denoising steps. Such a strategy is necessary to handle various scenarios effectively.

3. The proposed sparse attention acceleration is demonstrated to be efficient, as shown in Table 1.

4. The presentation is clear and easy to follow.

Weaknesses:

I do not have significant concerns. However, I am curious whether the observation that attention layers can be divided into solely spatial or temporal types is applicable to all diffusion models. For example, can the attention layers in multi-view diffusion models also be accelerated using the proposed method?

**Questions For Authors:**

NA

**Relation To Broader Scientific Literature:**

diffusion model acceleration

**Theoretical Claims:**

Yes

---

> ### Author Rebuttal · Authors · 2025-04-01
>
> Thank you for your valuable suggestions and questions. Below, we address each point raised.
>
> > Q1: Can the attention layers in multi-view diffusion models also be accelerated?
>
> Answer: Recent multi-view diffusion models such as MVDream [1] and CAT3D [2] have introduced 3D attention mechanisms to improve cross-view consistency and spatial coherence. MVDream takes an initial step in this direction but is limited by a relatively short context length (≤ 4k tokens), which diminishes the practical benefits of attention acceleration.
> CAT3D adopts finer-grained image representations and extends the attention span in 3D. It goes further than MVDream by applying 3D attention over longer sequences and finer feature maps. CAT3D also highlights the critical scalability issue: applying full 3D attention to high-resolution feature maps (e.g., 64×64) leads to prohibitively large sequence lengths—up to 32k tokens—making the approach computationally expensive. As a result, CAT3D restricts full 3D attention to lower-resolution feature maps (32×32 and below), where the overhead remains manageable.
>
> This design underscores the computational bottlenecks present in naive 3D attention and presents a clear opportunity for approaches like SVG.
> Since SVG can leverage the spatial and temporal redundancy, it could potentially enable efficient 3D full attention at larger resolutions, unlocking performance and scalability gains that current methods avoid due to cost.
> However, CAT3D does not open-source their weights. We are willing to validate the effectiveness of SVG on them when the weights are available.
>
> [1] Shi, Yichun, et al. "Mvdream: Multi-view diffusion for 3d generation." arXiv preprint arXiv:2308.16512 (2023).
>
> [2] Gao, Ruiqi, et al. "Cat3d: Create anything in 3d with multi-view diffusion models." arXiv preprint arXiv:2405.10314 (2024).

---

### Decision · Program_Chairs · 2025-05-01

**Decision:**

Accept (poster)

**Comment:**

The paper presents Sparse Video-Gen (SVG), a framework for accelerating video diffusion models by leveraging sparse attention patterns in Diffusion Transformers (DiTs). All reviewers acknowledge the method’s effectiveness in speeding up video generation while preserving high quality, with claims supported by detailed experimental results, including significant speedups and maintained PSNR values. The authors addressed the reviewers' initial concerns in their rebuttal adequately, providing additional technical details and video samples. Overall, the paper offers valuable contributions to efficient video generation.